# Cross-sectional study of the relationship between the spiritual wellbeing and psychological health among university Students

**Chi Hung Leung** [1]*, **Hok Ko Pong** [2]

**1** Department of Special Education & Counselling, The Education University of Hong Kong, Hong Kong, China, **2** Faculty of Management and Hospitality Technological and Higher Education Institute of Hong Kong, Hong Kong, China

* chhleung@eduhk.hk

## Abstract

University students' spiritual wellbeing has been shown to be associated with quality, satisfaction, and joy of life. This study tested the relationship between spiritual wellbeing and symptoms of psychological disorders (i.e., depression, anxiety and stress) among Chinese university students in Hong Kong. Cross-sectional data were collected from $N = 500$ students (aged 17–24; 279 women). The participants were asked to complete the Spiritual Health and Life-Orientation Measure (SHALOM) to evaluate the status of their spiritual wellbeing in the personal and communal, environmental, and transcendental domains, and the Depression, Anxiety and Stress Scale-21 (DASS-21) to assess their emotional states of depression, anxiety and stress. All domains of spiritual wellbeing were negatively associated with psychological distress. Hierarchical Multiple Regression showed that together the three domains of spirituality explained 79.9%, 71.3% and 85.5% of the variance in students' depression, anxiety and stress respectively. The personal and communal domain of spiritual wellbeing was the strongest predictor of psychological distress.

## Introduction

### Cross-sectional study of the relationship between the spiritual wellbeing and psychological health among university students

The drug and alcohol addiction, self-harm, and suicide were getting worse for college students, especially for those with family problems during the transition to college [1].The developmental transition from adolescence to emerging adulthood is one in which there can be considerable decreases in psychological health [2–4]. Compared with children and adult, adolescents are at increased risk of psychological disorders, such as anxiety and depression [3, 5, 6], and in recent decades they have been found to experience more stress, nervousness and vulnerability than adolescents in the past [7–9].

Mental health concerns are certainly evident among university students in Hong Kong. In the academic year 2016–2017, over 920 university students in Hong Kong sought counselling

**Data Availability Statement:** All relevant data are within the manuscript and its Supporting Information files.

**Funding:** This research is supported by the Departmental Research Fund (grant number

04883) received by LC-H. The funder had no role in study design, data collection and analysis, decision to publish, or preparation of the manuscript.

**Competing interests:** The authors have declared that no competing interests exist.

and other services for depression, anxiety and self-harm (South China Morning Post, 2016). In a sample of 1,119 university students in Hong Kong [10], self-reports showed that 68.5% had mild-to-severe symptoms of depression, often with co-morbid symptoms of anxiety. Moreover, 54.4% of the respondents indicated mild-to-severe anxiety symptoms, which were linked to academic difficulties. Furthermore, the rate of suicide has been increasing in this population. In Hong Kong, 52, 68 and 75 suicide cases amongst the continuously decreasing population of Chinese adolescents were reported in 2014, 2015 and 2016, respectively (Hong Kong Jockey Club Centre for Suicide Research and Prevention, 2018).

In the past decade there has been increasing research interest in the relationship between spirituality and psychological health [10, 11]. Spirituality, which is a concept related to spirit, immateriality, and metaphysics [12], comprises thoughts, dreams, visions, meaning, principles, and beliefs [13]. Furthermore, spirituality is not only regarded as pursuits of the aim, meaning, and purpose of life but it also may be either related with or separate from any religions [14]. Some studies have indicated that adolescents with higher spirituality had a higher quality of life in terms of their psychological health [15, 16] However, other studies have determined that no relationship exists between young adults' spirituality and psychological outcomes, such as anxiety [17, 18] The connection between spirituality and psychological health remains a controversial topic.

Two concepts that are related to spirituality are spiritual health and spiritual wellbeing. These two concepts themselves have overlapping but distinct meanings. The focus of the current study is spiritual wellbeing. However, because spiritual wellbeing is a relatively new area of research, much of the justification for our research questions is based on the literature on spiritual health. In the following sections we review studies on both constructs in relation to psychological health.

## Culture differences and spirituality

There is still a controversial argument for the relationship between spirituality and religion [19]. Spirituality is regarded as religion [20, 21] but Tanyi [22] and Palmer [13] still believe that spirituality is non-religious matters. In addition, spirituality could be illustrated differently in different cultures and traditions [23]. For example, Hofstede [24] and Triandis [25] found collectivist orientations in the Chinese, conversely individualistic orientations in the west. Chinese society is largely in favor of the perception of 'putting the interest of the whole above everything else' [26]. There is more emphasis on family values and harmonious relationships with people in Chinese cultures [27]. Also, there are pursuits of integration and adaptation of nature in Chinese cultures and traditions [28]. This idea is different from Western civilization's idea of conquering nature [29]. Lastly, transcendent spirituality refers to "God", "Divine" and "Creator" in the west [30], in contrast, transcendent spirituality in Chinese has a wide-ranging meaning that includes broad worldviews, such as any divine entity, ancients with great power, moral standards, higher spirits and higher self [31].

## Spiritual health and spiritual wellbeing

Spiritual health is a condition that guides people in identifying their meaning and purpose in life, as well as in enjoying love, happiness, peace and nature [32]. It represents the integration of the body, mind and spirit within an internally harmonious environment, and is based on connections with others, nature and the transcendent [33–35]. A spiritually healthy person experiences coherence and harmony of the mind and body [34, 36]. Spiritual health can also be conceptualized as a part of general health, along with physical and psychological health [33, 37–40].

Spiritual wellbeing is an indicator of spiritual health. An analogy might be that blood pressure is an indicator of physical health [12, 32]. Spiritual wellbeing has to do with an individual's current quality of life [12, 41], as well as the quality of current relations between one's emotions, thoughts and behaviour on the one hand and oneself, others, the transcendent and the environment on the other hand [12]. People with a strong spiritual wellbeing are characterized by current feelings of happiness, respect, contentment, forgiveness, mercy, humility, peace, beauty, honesty and harmony [42]; they possess a clear meaning and purpose of life and constantly engage in self-reflection and introspection to further improve themselves [43].

## Spiritual health and spiritual wellbeing in relation to depression

Depression involves complex emotional, cognitive and physical abnormalities. The emotional symptoms include continuous feelings of low mood, grief, emptiness, sorrow and frustration [44, 45]. The cognitive component includes a belief that one lacks value [44, 45], rumination, and thoughts of suicide [46, 47]. Physical symptoms include declines in energy and activity, insomnia, inattention and weight loss [46, 47]. Depression is also often accompanied by other psychological disorders and physical diseases [8, 48].

Even though the symptoms of depression are similar to characteristics of low spiritual health or low spiritual wellbeing, to date there has been no research on this topic. Importantly, the signs of depression are also features of a lack of spirituality [49]. Ganjavi [50] found that young people's development of spirituality significantly reduces their depression, anxiety and stress; whereas it increases their self-confidence. Koenig [51] and Ellison and Fannelly [52] found that spiritual development can effectively heal depression.

## Spiritual health and spiritual wellbeing in relation to anxiety

The word 'anxiety' is derived from the Latin *anxius*, meaning "suffocation," and refers to feelings of apprehension, worry or uneasiness [53]. When a person feels an invisible pressure for a long time, that feeling will result in inexplicable unhappiness, fear and confusion at rest or work [54]. Moreover, anxiety can make people experience restlessness or helplessness, or engage in problem behavior, such as restlessness and agitation [55].

Previous studies have shown a close relationship between spiritual wellbeing and anxiety. Fabbris et al. [56] found that nursing students with lower spiritual wellbeing had a higher chance of experiencing moderate or high anxiety, whereas the opposite was true for those with higher spiritual wellbeing. Boroumandzadeh and Sani [57] and Krause [58] reported that higher spiritual wellbeing was associated with lower anxiety and improve the overall health of students well. Anxious individuals are easily confused about the meaning of their own existence, feel a loss of trust in others, become disinterested in their surroundings and suspect others' beliefs and principles [59]. The sense of imminent danger often seen in anxiety [60, 61] is also an indicator of low spirituality [62].

## Spiritual health and spiritual wellbeing in relation to stress

Stress is a ubiquitous condition [63] that involves a physiological, cognitive and emotional response to pressures that are perceived to be unmanageable [64, 65]. In this state the person believes that they cannot achieve their goal [66]. Pressures that can elicit stress mainly originate from (1) external environmental, (2) organizational and (3) personal factors [67]. These factors are closely correlated and interact with one another. Given the same potential source of stress, each individual's perception differs [64]. That is, whether pressure creates stress depends entirely on the individual's subjective understanding of the situation [63].

Good and Willoughby [68] determined that the greater the stress on young people is, the greater the distortion of their spiritual health will be. Young adults facing high pressure experience a decline in spiritual wellbeing, shown in ways of their sense of pleasure, happiness index, the direction of life, their interpersonal relationships and their beliefs and values [69, 70].

## Dimensions of spiritual wellbeing

The literature just reviewed shows clearly that most research in this field has been on general spiritual health rather than the lived experience of spirituality in the form of spiritual wellbeing. Spiritual wellbeing is assumed to be a multidimensional construct, and this is reflected in measures used to assess it [71, 72]. Among the Chinese language measures, some assess one dimension of spiritual wellbeing, such as the Spiritual Transcendence Scale [73]. Others combine multiple aspects of spiritual wellbeing under one global score, such as the Daily Spiritual Experience Scale [74] or under one subscale score, for example the spiritual subscale in the Chinese Positive Youth Development Scale [75], a measure modelled after the Own Purpose in Life Questionnaire. Only one measure includes multiple distinct subscales to represent multiple domains of spiritual wellbeing. Specifically, the Chinese version of the Spiritual Health and Life-Orientation Measure (SHALOM) [76] has subscale scores for the personal and communal, environmental, and transcendent domains of spiritual wellbeing.

## Potential response to the campus mental health crisis

Religion is found to contribute to the mental well-being of people in large-scale studies and in-depth research [77, 78]. Besides, the study of Astin et al. [79] found that frequent participation in self-reflection activities, such as yoga, meditation and mindfulness programs, contributed to spiritual growth and inner peace of university students. Fenton et al. [80] indicated these programs could help reduce perceived stress, anxiety, depression, and negative mood of college students. Studying abroad, leadership camp training, cultural interaction between different races, inter-disciplinary learning and community service activities are also the essential actions that affect students' character cultivation [79].

Physical exercises [81] and music training [81, 82] are beneficial to the development of college students' well-being. Numerous research, such as [83] and [82] respectively have determined that people who regularly perform physical exercises and make use of music can certainly keep their joyful manner, positivity, self-assurance and addiction control.

## Adoption of SHALOM in the Chinese youth

Fisher initially developed the Spiritual Health and Life-Orientation Measure (SHALOM) in 1998 [12]. The 20-item form of SHALOM has now been popular and translated into 29 languages, also it has been always adopted in numerous research with higher consistent reliability [12, 84–88]. SHALOM was adopted in the various studies in the west, including Hebrew [89], Persian [90] and Brazilian–Portuguese populations [91, 92], the reliable four-factor structure (i.e. personal, communal, environmental and transcendental domains) was found in 'lived experience' and 'ideal value' subscale. The three-factor structure of SHALOM would be identified in the Chinese population because of the cultivation of Confucian thoughts [93]. The differences in the structure of spirituality partially reveal cultural variation [31]. Fisher [94] modified the transcendent section of SHALOM to have broad coverage for, including God or a divine, deceased person, higher power and higher self, amongst others. With the use of the SHALOM scale, a significant difference in the transcendental spiritual wellbeing was also found between religious and non-religious youths in the Chinese [93] and the west population [92].

The 20-item Chinese translation of the Spiritual Health and Life-orientation Measure (SHALOM) [31] is not only relatively and satisfactorily complete to cover various aspects of spirituality, including the individual's sense of connectedness to oneself, others, the natural world and transcendence [95, 96]. But it also was validated as a reliable measure of spiritual health amongst youth Chinese [93]. This is the measure of spiritual wellbeing used in the current study.

## Present study

The current study was designed to test relations between spiritual wellbeing and psychological health. This research moves beyond the current literature on this topic by examining the predictive value of multiple dimensions of spiritual wellbeing. Accordingly, this study addressed the following research questions:

1. Is spiritual wellbeing associated with lower psychological distress (i.e., lower depression, anxiety and stress) among Chinese university students?

2. Are these associations equally strong for different types of psychological distress (depression, anxiety, and stress)?

3. Are these associations equally strong for different domains of spiritual wellbeing (personal/communal, environmental, and transcendental) among Chinese university students?

## Method

Approval to conduct this study was obtained from the Human Research Ethics Committee of the Education University of Hong Kong.

## Participants

There were 500 Chinese university students who participated either in 2018 ($n$ = 250) or 2019 ($n$ = 250) at two universities in Hong Kong, respectively. These participants responded to an invitation that was sent to 800 students, constituting a response rate of 62.5%. A power analysis of the sample size with a confidence level of 95% is 383 [97]. A sample of 500 Chinese university students is enough for the present study. The result of the power indicated a precise estimate of the regression coefficient with a small confidence interval around it and the representativeness of the sample.

In 2018, the first sample consisted of 250 Chinese university students (131 women, 119 men; aged 17–23). Most participants reported no religious beliefs (59.2%), with the rest being Other Christians (30%), Catholics (5.2%), Buddhists (13; 4%), and Taoists (1.6%). In 2019, the second sample also consisted of 250 Chinese university students (148 women, 102 men; aged 17–23). Most participants reported no religious beliefs (52%), with the rest being Other Christians (35.2%), Catholics (6%), Buddhists (4.8%), and Taoists (2%).

A series of t-tests found no significant difference between the 2018 and 2019 samples in the mean scores on any of the spiritual wellbeing domains, depression, anxiety or stress ($ps$ > .05). Thus, the samples from 2018 and 2019 were combined in the analyses.

## Measures

**Demographics.** Participants provided demographic information about gender, age, major, year in school, whether or not they held religious beliefs, and religious affiliation.

**Spiritual wellbeing.** The Spiritual Health and Life-Orientation Measure (SHALOM) assesses the spiritual wellbeing in four domains [12]. In the current study we used a 20-item

version of the SHALOM that was developed for use in the Chinese cultural context [93]. The measure assesses spiritual wellbeing in the personal and communal (10 items), environmental (5 items), and transcendental (5 items) domains. Example items include "Joy in life," "Love of other people," "Peace with the Transcendent," and "Harmony with the environment." The participants rated the importance of each item with regard to its presence in their daily life (also called lived experience). Items were rated using a five-point Likert scale ranging from 1 (very low) to 5 (very high). The domain scores are the mean of domain item scores. The Cronbach alphas for the personal and communal domain, environmental domain, and transcendental domain were .88, .82, and .95, respectively.

The original SHALOM included four domains, namely personal, communal, environmental, and transcendental [33]. Subsequent research in the Chinese context supported a three-domain model that combined the personal and communal domains, based on factor analysis [93] as well as evidence of a close relation between the meanings of the personal and communal domains in Chinese culture [24]. In the current study Principal Components Analysis showed that the three-domain version was appropriate for use in our sample (Kaiser–Meyer–Olkin value = 0.92; Bartlett's test of sphericity significant at $p < .001$). Exploratory Factor Analysis identified three factors (each with an eigenvalue > 1.0) corresponding to the personal and communal, environmental, and transcendental domains, which explained 35.29%, 16.56%, and 9.00% of the variance respectively. Table 1 shows the factor loadings for the three-factor model.

**Table 1. Results of Exploratory Factor Analysis of the SHALOM items (N = 500).**

|  | Component | | |
|---|---|---|---|
|  | **Personal–Communal** | **Environmental** | **Transcendental** |
| Q1: Love for other people | **0.612** | 0.197 | 0.018 |
| Q2: Personal relationship with the Divine | 0.125 | **0.873** | 0.074 |
| Q3: Forgiveness towards others | **0.631** | 0.128 | 0.124 |
| Q4: Connection with nature | 0.193 | 0.054 | **0.776** |
| Q5: Sense of identity | **0.554** | 0.001 | 0.246 |
| Q6: Worship of the Creator | 0.101 | **0.860** | 0.191 |
| Q7: Appreciation of breathtaking views | 0.081 | 0.350 | **0.612** |
| Q8: Trust amongst individuals | **0.700** | 0.029 | 0.172 |
| Q9: Self-awareness | **0.613** | −0.018 | 0.277 |
| Q10: Oneness with nature | 0.165 | 0.075 | **0.841** |
| Q11: Oneness with God | 0.125 | **0.903** | 0.152 |
| Q12: Harmony with the environment | 0.353 | 0.152 | **0.638** |
| Q13: Peace with God | 0.094 | **0.922** | 0.131 |
| Q14: Joy in life | **0.649** | 0.329 | 0.197 |
| Q15: Prayer in life | 0.123 | **0.891** | 0.073 |
| Q16: Inner peace | **0.718** | 0.084 | 0.103 |
| Q17: Respect for others | **0.764** | 0.059 | 0.128 |
| Q18: Meaning in life | **0.746** | 0.115 | 0.163 |
| Q19: Kindness towards other people | **0.744** | 0.047 | 0.124 |
| Q20: Sense of 'magic' in the environment | 0.232 | 0.107 | **0.701** |
| Explanation of variance for each factor (%) | **35.29** | **16.56** | **9.00** |
| Cumulative variance (%) | **35.29** | **51.85** | **60.85** |

Note: Items loaded on each factor are in boldface.

**Psychological health.**   The Chinese version [98] of the Depression, Anxiety, and Stress Scale [99, 100] was used to measure psychological health in three areas, namely depression, anxiety and stress (seven items each). The participants used a four-point scale ranging from 0 (did not apply to me at all) to 3 (applied to me very much or most of the time) to rate the extent to which they experienced a list of 21 statements over the last week. The DASS-21 has been shown to have high reliability as well as strong convergent and discriminant validity [99, 100]. Possible scores range from 0 to 21 in each of the three areas, with higher scores indicating higher distress. Cronbach's alphas for depression, anxiety, and stress were 0.91, 0.84, and 0.90 in the normative sample, respectively [101, 102].

To provide clinically meaningful information about the psychological health of the students in our sample, we converted the DASS-21 scores to their equivalent on the42 items of DASS by multiplying each item score by two. The new scores could then be categorized using five severity labels: normal, mild, moderate, severe, and extremely severe [100].

**Procedure.**   Approval to conduct this study was obtained from the Research Ethics Committee of the institutions. Potential participants were told the purpose of the study and were informed that participation was voluntarily, that they could withdraw at any time without penalty or prejudice, that no compensation was offered, and that all information was confidential. The volunteers provided written informed consent to participate. Under the bilingual cultural context of Hong Kong, the questionnaires were presented in Chinese and English and participants could complete either version. The participants were given the paper questionnaires and completed them with pencils in the classroom and then returned to responsible teachers.

## Results

SPSS Version 23 was adopted for data analysis. Data cleaning was performed to detect any missing values, coding errors, or illogical data values.

### Descriptive statistics

Table 2 shows the descriptive statistics. As a group, the university students had average scores on the three SHALOM domains and on the three DASS-21 subscales. Table 3 shows that 43.2%, 38.4%, and 35.6% reported symptoms of mild depression, anxiety, and stress, respectively; whereas 2%, 3%, and 1.2% indicated symptoms of moderate depression, anxiety, and stress, respectively. On the basis of the score ranges from the DASS-21 manual, the mean scores of depression for all students were found at nearly the mild level (8.90 ± 2.61), the mean anxiety scores were at a normal level (6.53 ± 1.77), and the mean stress scores were also at a normal level (12.94 ± 3.84).

### Quantitative findings

Table 4 showed that relations involving the personal and communal domain and the environmental domains were moderate to strong (Pearson's *r* values from -.66 to -.80) whereas relations involving the transcendental domain were moderate (-.49 to -.53). Hierarchical Multiple Regression showed that together the three domains of spirituality explained 79.9%, 71.3% and 85.5% of the variance in students' depression, anxiety and stress respectively shown in Table 5.

### Statistical analysis

The quantitative data showed a statistically significant negative relationship between students' spiritual wellbeing in all domains (i.e., personal and communal, environmental, and transcendental) and symptoms of the three psychological disorders (i.e., depression, anxiety, and

**Table 2. Descriptive statistics: Participants' demographics and their relationship with depression, anxiety and stress and spiritual wellbeing (N = 500).**

| Factors | N (%) | Depression, mean (SD) | Anxiety, mean (SD) | Stress, mean (SD) | Spiritual wellbeing (personal and communal) | Spiritual wellbeing (environmental) | Spiritual wellbeing (transcendental) |
|---|---|---|---|---|---|---|---|
| All | 500 (100.0) | 8.90 (2.61) | 6.53 (1.77) | 12.94 (3.84) | 4.08 (0.51) | 3.53 (0.68) | 3.11 (1.06) |
| Years of samples collected in | | | | | | | |
| 1. 2018 in University A | 250 | 8.94 (2.66) | 6.51 (1.79) | 12.99 (3.86) | 4.064 (0.53) | 3.56 (0.68) | 3.09 (1.08) |
| 2. 2019 in University B | 250 | 8.85 (2.56) | 6.55 (1.75) | 12.90 (3.83) | 4.090 (0.50) | 3.51 (0.67) | 3.13 (1.05) |
| | | $t = 0.411$ | $t = -0.253$ | $t = 0.279$ | $t = -0.558$ | $t = 0.779$ | $t = -0.437$ |
| Age | | | | | | | |
| (19.94 ± 1.14) | | | | | | | |
| 1. 17–19 | 138 (27.6) | 8.87 (2.64) | 6.60 (1.66) | 13.33 (3.76) | 4.04 (0.55) | 3.53 (0.74) | 3.23 (1.06) |
| 2. 20 | 226 (45.2) | 8.84 (2.62) | 6.43 (1.74) | 12.82 (3.90) | 4.10 (0.52) | 3.52 (0.67) | 3.07 (1.08) |
| 3. 21–23 | 136 (27.2) | 8.02 (2.47) | 6.01 (1.98) | 11.95 (4.12) | 4.20 (0.41) | 3.68 (0.70) | 3.47 (1.05) |
| | | $F_{(2, 499)} = 1.384$ | $F_{(2, 499)} = 1.245$ | $F_{(2, 499)} = 1.124$ | $F_{(2, 499)} = 0.098$ | $F_{(2, 499)} = 0.413$ | $F_{(2, 499)} = 1.45352$ |
| Gender | | | | | | | |
| 1. Male | 221 (44.2) | 8.91 (2.75) | 6.55 (1.81) | 12.78 (3.98) | 4.09 (0.52) | 3.54 (0.67) | 3.11 (1.07) |
| 2. Female | 279 (55.8) | 8.89 (2.50) | 6.52 (1.74) | 13.08 (3.73) | 4.07 (0.51) | 3.52 (0.69) | 3.10 (1.06) |
| | | $t = 0.068$ | $t = 0.23$ | $t = -0.859$ | $t = 0.532$ | $t = 0.281$ | $t = 0.096$ |
| Religious beliefs | | | | | | | |
| - No | 271 (54.2) | 9.47 (2.49) | 6.92 (1.74) | 13.78 (3.57) | 4.04 (0.50) | 3.47 (0.67) | 2.51 (0.90) |
| - Yes | 229 (55.8) | 8.22 (2.60) | 6.07 (1.69) | 11.96 (3.91) | 4.12 (0.52) | 3.60 (0.68) | 3.82 (0.76) |
| | | $t = 5.491^{**}$ | $t = 5.532^{**}$ | $t = 5.437^{**}$ | $t = -1.740$ | $t = -2.110$ | $t = -17.443^{*}$ |
| Religious affiliation | | | | | | | |
| 1. None | 278 (55.6) | 9.45 (2.47) | 6.91 (1.72) | 13.79 (3.54) | 4.04 (0.50) | 3.47 (0.67) | 2.52 (0.89) |
| 2. Other Christian | 163 (32.6) | 8.25 (2.58) | 6.04 (1.73) | 11.90 (3.98) | 4.11 (0.51) | 3.57 (0.67) | 3.92 (0.71) |
| 3. Catholic | 28 (5.6) | 7.71 (2.71) | 6.00 (1.44) | 11.21 (3.75) | 4.14 (0.62) | 3.86 (0.57) | 3.94 (0.77) |
| 4. Buddhist | 22 (4.4) | 8.36 (2.66) | 6.09 (1.90) | 12.55 (3.76) | 4.22 (0.48) | 3.50 (0.61) | 3.34 (0.86) |
| 5. Taoist | 9 (1.8) | 8.44 (3.28) | 6.44 (1.94) | 12.00 (4.69) | 4.12 (0.52) | 3.73 (1.02) | 3.44 (0.54) |
| | | $F_{(4, 499)} = 7.840^{**}$; | $F_{(4, 499)} = 7.830^{**}$; | $F_{(4, 499)} = 8.486^{**}$; | $F_{(4, 499)} = 1.256$ | $F_{(4, 499)} = 2.450$ | $F_{(4, 499)} = 82.713^{**}$; |
| | | 2 > 1, 3 > 1 | 2 > 1, 3 > 1, 4 > 1 | 2 > 1, 3 > 1 | | | 2 > 1, 3 > 1, 4 > 1, |
| | | | | | | | 5 > 1, 2 > 4, 3 > 4, |
| Academic major disciplines | | | | | | | |
| 1. Arts and humanities | | | | | | | |
| 2. Business | 125 (25) | 8.82 (2.34) | 6.50 (1.62) | 13.06 (3.56) | 4.09 (0.49) | 3.50 (0.72) | 3.11 (1.07) |
| 3. Science | 137 (27.4) | 8.91 (2.63) | 6.57 (1.71) | 12.99 (3.96) | 4.07 (0.53) | 3.51 (0.72) | 3.05 (1.07) |

*(Continued)*

**Table 2.** (Continued)

| Factors | N (%) | Depression, mean (SD) | Anxiety, mean (SD) | Stress, mean (SD) | Spiritual wellbeing (personal and communal) | Spiritual wellbeing (environmental) | Spiritual wellbeing (transcendental) |
|---|---|---|---|---|---|---|---|
| 4. Social science | 142 (28.4 | 8.99 (2.47) | 6.59 (1.70) | 12.92 (3.69) | 4.07 (0.49) | 3.56 (0.60) | 3.19 (1.05) |
| | 96 (19.2) | 8.85 (3.09) | 6.44 (2.12) | 12.77 (4.26) | 4.08 (0.55) | 3.57 (0.68) | 3.08 (1.07) |
| | | $F(3, 499) = 0.104$ | $F(3, 499) = 0.182$ | $F(3, 499) = 0.110$ | $F(3, 499) = 0.081$ | $F(3, 499) = 0.368$ | $F(3, 499) = 0.418$ |
| Years of study | | | | | | | |
| 1. Year 1 | 117 (23.4) | 8.96 (2.61) | 6.50 (1.68) | 13.08 (3.81) | 4.04 (0.51) | 3.50 (0.66) | 3.09 (1.12) |
| 2. Year 2 | 184 (36.8) | 9.11 (2.63) | 6.71(1.76) | 13.11 (3.79) | 4.07 (0.50) | 3.52 (0.71) | 3.01 (1.03) |
| 3. Year 3 | 145 (29) | 8.72 (2.61) | 6.41(1.79) | 12.74 (3.96) | 4.12 (0.54) | 3.56 (0.65) | 3.17 (1.05) |
| 4. Year 4 | 54 (10.8) | 8.52 (2.52) | 6.33(1.89) | 12.63 (3.80) | 4.11 (0.49) | 3.59 (0.68) | 3.33 (1.07) |
| | | $F(3, 499) = 1.032$ | $F(3, 499) = 1.059$ | $F(3, 499) = 0.409$ | $F(3, 499) = 0.410$ | $F(3, 499) = 0.789$ | $F(3, 499) = 0.215$ |

$^{**}p < .01$;

$^{***}p < .001$

stress). The findings indicated that the lower the spiritual wellbeing score on each domain, the higher the score on each psychological disorder subscale.

We next performed a series of analyses (Table 2) examining potential differences in psychological disorders (i.e., depression, anxiety, and stress) and spiritual wellbeing domains (i.e., personal and communal, environmental, and transcendental) based on the various demographic characteristics. One-way ANOVAs revealed no significant differences in the average score for depression, anxiety, stress, or spiritual wellbeing status in any of the specific domains across the participants' ages, academic major disciplines, and years of study. Similarly, a series of t-tests also found no significant differences in those same scores based on the year that the sample data were collected or based on participant gender.

**Table 3. Criterion scores and summary of participants for DASS levels of severity of depression, anxiety, and stress.**

| | Depression | Anxiety | Stress |
|---|---|---|---|
| | Frequency (%) | Frequency (%) | |
| Normal | 0–9 | 0–7 | 0–14 |
| | 274 (54.8%) | 293 (58.6%) | 316 (63.2%) |
| Mild | 10–13 | 8–9 | 15–18 |
| | 216 (43.2%) | 192 (38.4%) | 178 (35.6%) |
| Moderate | 14–20 | 10–14 | 19–25 |
| | 10 (2%) | 15 (3%) | 6 (1.2%) |
| Severe | 21–27 | 15–19 | 26–33 |
| | 0 (0%) | 0 (0%) | 0 (0%) |
| Extremely Severe | 28+ | 20+ | 34+ |
| | 0 (0%) | 0 (0%) | 0 (0%) |

**Source:** [100]

**Table 4. Pearson correlations between three domains of spiritual wellbeing and three types of psychological disorder among undergraduate students (N = 500).**

|  | Personal and Communal | Environmental | Transcendental |
|---|---|---|---|
| Depression | -0.770** | −0.709** | −0.534** |
| Anxiety | -0.739** | −0.662** | −0.492** |
| Stress | -0.795** | −0.754** | −0.525** |

$^*p < .05$

$^{**}p < 0.01 {}^{***}p < .001$

In regard to demographic questions about religion, a series of t-tests found a significant difference between students who did and did not report having religious beliefs on the mean scores of depression, anxiety, stress, and the transcendental domain of spiritual wellbeing. Participants with religious beliefs have higher scores in DASS 21 and spiritual wellbeing than

**Table 5. Results of hierarchical regression analyses with spiritual wellbeing (SWB) in the personal and communal, environmental, and transcendental domains as predictors of participants' depression, anxiety, and stress.**

| Variable | | $\beta$ | $T$ | $F$ | $R$ | $R2$ | $\Delta R2$ | Adjusted $R2$ |
|---|---|---|---|---|---|---|---|---|
| **Depression** | | | | | | | | |
| Step 1 | | | | 726.994 | 0.770 | 0.593 | 0.593 | 0.593 |
| | Personal and Communal SWB | −0.770 | −26.963 | | | | | |
| Step 2 | | | | 705.377 | 0.860 | 0.739 | 0.146 | 0.738 |
| | Personal and Communal SWB | −0.557 | −21.238 | | | | | |
| | Environmental SWB | −0.438 | −16.690 | | | | | |
| Step 3 | | | | 655.216 | 0.894 | 0.799 | 0.059 | 0.797 |
| | Personal and Communal SWB | −0.515 | −22.081 | | | | | |
| | Environmental SWB | −0.369 | −15.538 | | | | | |
| | Transcendental SWB | −0.261 | −12.054 | | | | | |
| **Anxiety** | | $\beta$ | $T$ | $F$ | $R$ | $R2$ | $\Delta R2$ | Adjusted $R2$ |
| Step 1 | | | | 598.644 | 0.739 | 0.546 | 0.546 | 0.545 |
| | Personal and Communal SWB | −0.739 | −24.467 | | | | | |
| Step 2 | | | | 494.441 | 0.816 | 0.666 | 0.120 | 0.664 |
| | Personal and Communal SWB | −0.546 | −18.364 | | | | | |
| | Environmental SWB | −0.396 | −13.333 | | | | | |
| Step 3 | | | | 409.778 | 0.844 | 0.713 | 0.047 | 0.711 |
| | Personal and Communal SWB | −0.509 | −18.242 | | | | | |
| | Environmental SWB | −0.335 | −11.806 | | | | | |
| | Transcendental SWB | −0.233 | −9.005 | | | | | |
| **Stress** | | $\beta$ | $T$ | $F$ | $R$ | R2 | $\Delta R2$ | Adjusted R2 |
| Step 1 | | | | 856.238 | 0.795 | 0.632 | 0.632 | 0.632 |
| | Personal and Communal SWB | −0.795 | −29.262 | | | | | |
| Step 2 | | | | 1044.378 | 0.899 | 0.808 | 0.176 | 0.807 |
| | Personal and Communal SWB | −0.561 | −24.912 | | | | | |
| | Environmental SWB | −0.480 | −21.304 | | | | | |
| Step 3 | | | | 973.912 | 0.925 | 0.855 | 0.047 | 0.854 |
| | Personal and Communal SWB | −0.524 | −26.455 | | | | | |
| | Environmental SWB | −0.419 | −20.762 | | | | | |
| | Transcendental SWB | −0.233 | −12.685 | | | | | |

participants without religions. Across religious affiliations, a series of one-way ANOVAs showed significant differences in the mean scores for depression, anxiety, stress, and the transcendental domain of spiritual wellbeing. Post-hoc analyses using the LSD test indicated significant differences between Other Christians and Catholics in the mean scores for depression, anxiety, stress, and the transcendental domain of spiritual wellbeing ($p < 0.05$).

Next, we performed hierarchical regression analyses using the spiritual wellbeing domains as predictor variables and the depression, anxiety and stress scores as dependent variables in three analyses (Table 5). For depression, students' personal and communal domain of spiritual wellbeing was entered into the equation in Step 1, $F(1, 498) = 726.994$, $p < 0.001$, accounting for 59.3% of the variance in depression. In Step 2, the environmental domain was entered into the equation, $F(2, 497) = 705.377$, $p < 0.001$. After Step 2, 73.9% of the variance in depression was accounted for, an additional 14.6%. In Step 3, the transcendental domain was entered into the equation, $F(3, 496) = 655.216$, $p < 0.001$. After Step 3, 79.9% of the variance in depression was accounted for, an additional 5.9% of the variance.

For anxiety, in Step 1, students' personal and communal domain of spiritual wellbeing was entered into the equation, $F(1, 498) = 598.644$, $p < 0.001$, accounting for 54.6% of the variance in anxiety. In Step 2, the environmental domain was entered into the equation, $F(2, 497) = 494.441$, $p < 0.001$. After Step 2, 66.6% of the variance in anxiety was accounted for, an additional 12% of the variance, which is excessively large. In Step 3, the transcendental domain was entered into the equation, $F(3, 496) = 409.788$, $p < 0.001$. After Step 3, 71.3% of the variance in anxiety was accounted for, an additional 4.7% of the variance.

For stress, in Step 1, students' personal and communal domain of spiritual wellbeing was entered into the equation, $F(1, 498) = 856.238$, $p < 0.001$, accounting for 63.2% of the variance in stress. In Step 2, the environmental domain was entered into the equation, $F(2, 497) = 1044.378$, $p < 0.001$. After Step 2, 80.8% of the variance in stress was accounted for, an additional 17.6% of the variance, which is extremely large. In Step 3, the transcendental domain was entered into the equation, $F(3, 496) = 973.912$, $p < 0.001$. After Step 3, 85.5% of the variance in stress was accounted for, an additional 4.7% of the variance, which is relatively small.

## Discussion

University students are going through substantial transformations from the life stage of adolescence to emerging adulthood [4]. In the current study we examined whether spiritual wellbeing was associated with better psychological health during this period. Chinese undergraduate students were asked about their depression, anxiety and stress levels, as well as their experienced spiritual wellbeing in the personal and communal, environmental and transcendental domains. The results of this cross-sectional study showed negative and statistically significant relationships between symptoms of psychological distress and the multiple dimensions of spiritual wellbeing. In the Asia-Pacific region, it is the only study to contribute the links between spiritual wellbeing and psychological disorders among university students.

In this sample the mean scores for depression, anxiety and stress were all considered normal, although some students' distress fell in the mild range (43.2%, 38.4% and 35.6%, respectively). These rates are somewhat lower than those found in university samples using similar measures in Malaysia [101] and Turkey [7]. Importantly, however, even given a perhaps restricted range of psychological symptoms, students who reported higher-than-average distress also reported showed significantly lower spiritual wellbeing. This suggests that psychological intervention and spiritual counselling may both be relevant components of support provided to university students in psychological distress.

## Spiritual wellbeing and depression

The findings showed that students with lower levels of spiritual wellbeing reported higher levels of depression, and this association was found for all domains of spiritual wellbeing (personal and communal, environmental, transcendent). These results are consistent with other evidence that university students with symptoms of depression had lower spiritual wellbeing in the personal and communal domains [62]. Another study found that lower spiritual wellbeing was associated with greater sadness among university students. The results are also consistent with evidence that people with more connection with nature have a lower risk of being depressed [103, 104]; there is even longitudinal evidence that more contact with nature in childhood is strongly linked to fewer symptoms of depression in adulthood [103]. Townsend and Weerasuriya [105] also found that a connection with nature was effective in reducing insomnia and depression. However, the present findings were not consistent with research showing a small but significant positive association between transcendental spiritual wellbeing and depression [106].

## Spiritual wellbeing and anxiety

The findings showed students with lower levels of spiritual wellbeing, across all domains, reported higher levels of anxiety. The findings were consistent with research by Jafari [107], who showed that university students with symptoms of anxiety exhibited lower spiritual wellbeing in the personal and communal domains. There are clear conceptual links between spiritual wellbeing and anxiety. For example, students with low spiritual wellbeing may worry about the meaning and purpose of their life, and they often feel uncomfortable with undefined, unfamiliar or vague sensations [59]. By contrast, university students with high spiritual wellbeing have clear goals and meaning of life, and they often feel peaceful and stable, even under uncertainties [42].

Moreover, the findings with regard to the environmental domain of spiritual wellbeing are supported by Maller's [108] research, which indicated that returning and being close to nature can calm unstable feelings and nervousness. Activities in nature (e.g., hiking) can not only increase bodily strength, willpower and the ability to release daily pressures [109], but they can also promote psychological health and prevent emotional instability and anxiety [110]. However, the findings of this study are inconsistent with those of Negi et al. [62], who showed a significantly positive relationship between the transcendental domain of spiritual wellbeing and anxiety. The differences in the findings of the studies might be caused by the combination of age groups, different measures utilized and conceptualization of constructs employed.

## Spiritual wellbeing and stress

The findings showed that students with lower levels of spiritual wellbeing, across domains, reported higher levels of stress. In another study of university students, Lee [111] also found a significant negative correlation between stress symptoms and spiritual wellbeing in the personal and communal domains. The results were also consistent with those of Negi et al. [62]. Schafer [112] indicated that college students with a clear meaning or purpose in life had less personal distress than those with a vague or confusion of life. Debnam et al. [113] found that high stress among youth was significantly associated with the intake of alcohol, tobacco and other drugs, which distort their spiritual wellbeing in the personal and communal domains.

The findings of this study on the relationship between transcendental spiritual wellbeing and stress do not coincide with the results of O'Connor [114], who revealed no association, and with those of Negi et al. [62], who showed a positive relationship. O'Connor [114] explained that the lack of a relationship between transcendental spiritual wellbeing and

psychological distress may be accounted for by the lack of sensitivity of the religion assessment used. Negi et al. [62] indicated that there were struggles and contradictions between the pursuit of religious ideal and consideration of reality. People with higher spiritual wellbeing in the transcendental domain may have more shackles and restrictions in daily life.

## Limitations

The study has four major limitations. Firstly, the generalizability of the findings may be limited because the sample of 500 students from two universities is relatively small compared with the total population of students in Hong Kong. Secondly, although the SHALOM has been shown to be a reliable and valid multi-dimensional measure of spiritual wellbeing, there is still a lack of agreement in the literature about the meaning of terms and concepts such as spirituality, spiritual wellbeing and transcendence. Thirdly, with regard to the DASS-21, participants might not be accurate reporters on their own distress, and the measure does not include a validity scale to check for inconsistent answers. Fourthly, all information was based on self-report questionnaires, raising concern about inflated correlations due to shared method variance.

## Conclusion

The findings of this study suggest that university students with high spiritual wellbeing are also likely to experience fewer symptoms of depression, anxiety and stress. All domains of spiritual wellbeing (personal and communal, environmental, and transcendental) were associated with lower psychological distress, but the link was especially strong for the personal and communal domain. This is the first study on the spiritual wellbeing of university students in the Asia-Pacific region. The results have implications for incorporating spiritual counselling into interventions for students experiencing psychological problems.

## Supporting information

**S1 Data.**
(SAV)

**S1 Table. Results of Exploratory Factor Analysis of the SHALOM items (N = 500).**
(DOCX)

**S2 Table. Descriptive statistics: Participants' demographics and their relationship with depression, anxiety and stress and spiritual wellbeing (N = 500).**
(DOCX)

**S3 Table. Criterion scores and summary of participants for DASS levels of severity of depression, anxiety, and stress.**
(DOCX)

**S4 Table. Pearson correlations between three domains of spiritual wellbeing and three types of psychological disorder among undergraduate students (N = 500).**
(DOCX)

**S5 Table. Results of hierarchical regression analyses with Spiritual Wellbeing (SWB) in the personal and communal, environmental, and transcendental domains as predictors of participants' depression, anxiety, and stress.**
(DOCX)

## Author Contributions

**Formal analysis:** Chi Hung Leung.

**Writing – original draft:** Chi Hung Leung, Hok Ko Pong.

**Writing – review & editing:** Chi Hung Leung, Hok Ko Pong.

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
