## [Decision Letter · Decision Letter 0]

19 Oct 2020

PONE-D-20-25063

Cross-Sectional Study of the Relationship between the Spiritual Wellbeing and Psychological Health among University Students

PLOS ONE

Dear Dr. Chi Hung Leung and co-workers,

Thank you for submitting your manuscript to PLOS ONE. After careful consideration, we feel that it has merit but does not fully meet PLOS ONE’s publication criteria as it currently stands. Therefore, we invite you to submit a revised version of the manuscript that addresses the points raised during the review process.

Two items on the subject (more background on the SHALOM measure, a synopsis of the current literature on students' well-being and spirituality) and two minor issues on the statistics (add a power-analysis, combine the statistical methods in a section satisitical analysis). 

We look forward to receiving your revised manuscript.

Kind regards,

Jacobus P. van Wouwe, MD PhD

Academic Editor

PLOS ONE

Additional Editor Comments:

Your well written manuscript on a well performed study focuses on a relevant subject. Your findings make a nice contribution on the subject.

Two issues on the content: more emphasis on the SHALOM measures and its modifications and some more general context on spirituality among university students will both add to the impact of your manuscript. Reviewer #1 gives some valuable suggestions.

The statistical analysis is appropriate, a power analysis will explain your sample size and the statistical analysis could be combined in a separate section of your methods.

Journal Requirements:

"NIL"

"NIL"

5. Please include your tables as part of your main manuscript and remove the individual files. Please note that supplementary tables (should remain/ be uploaded) as separate "supporting information" files.

Reviewers' comments:

Reviewer's Responses to Questions

**Comments to the Author**

1. Is the manuscript technically sound, and do the data support the conclusions?

Reviewer #1: Yes

Reviewer #2: Partly

2. Has the statistical analysis been performed appropriately and rigorously? 

Reviewer #1: Yes

Reviewer #2: No

3. Have the authors made all data underlying the findings in their manuscript fully available?

Reviewer #1: No

Reviewer #2: Yes

4. Is the manuscript presented in an intelligible fashion and written in standard English?

Reviewer #1: Yes

Reviewer #2: Yes

5. Review Comments to the Author

Reviewer #1: This is a potentially useful contribution to the literature on college students, spirituality, and psychological outcomes, but it needs a more sophisticated and nuanced framing of spirituality (the central independent variable).

The authors write about spirituality in the same way a scholar might write about temperature (you can be high or low). This neglects the contested nature of spirituality and cultural differences in how spirituality is defined. The authors do briefly discuss the use of a modified SHALOM measure that has been adapted to Hong Kong. This is a missed opportunity to talk more about cross-cultural differences in spirituality.

To improve the paper, the author's should make this a paper about the SHALOM measure, rather than a general paper on how spirituality improves mental health. It is simply too large a claim to say spirituality is good for your health. Does all spirituality look the same? What about spirituality that might be attached to harmful ideas (a "morbid scrupulosity" in Catholicism, for example)? By narrowing the focus of the paper to SHALOM, the authors could avoid a lot of these problems.

The paper should be revised to include a section explaining the rationale for using SHALOM and the history of this concept (for example, the article in Religions by J. Fisher). Readers would benefit from a bit more of the back story of this item and why it is a good measure to use with college students. As it is written now, the paper assumes readers know about SHALOM. The literature review should justify why SHALOM is a good approach to use.

The paper should also say a bit more about the mental health crisis among college students, as well as efforts of religious professionals to respond to it (see Varun Soni's work, for example at USC). Mindfulness programs, yoga and meditation, as well as chaplaincy, have been used to respond to the campus mental health crisis. People in all of these programs would be interested in this paper. Finally, the paper should draw on the research of UCLA scholars Alexander Astin, Helen Astin, and Jennifer Lindholm, the most cited figures in U.S. research on spirituality and higher education, missing from this paper so far.

If these changes are made, this will be a much stronger paper. The findings seem sound. The framing is the area for improvement.

Reviewer #2: The paper is very well written, the statistics performed were fairly routine and the results follow from the analyses. Specifically, the findings of this study suggest that university students with high spiritual wellbeing are also likely to experience fewer symptoms of depression, anxiety and stress. All domains of spiritual wellbeing (personal and communal, environmental, and transcendental) were, in fact, associated with lower psychological distress.

There are two concerns.

1. The investigators note that Cross-sectional data was collected from 500 students. Although the work is mainly descriptive, analysis and p-values are given throughout. There is no statistical power motivation for the sample size. This should be addressed by the investigators.

2. On a minor note, the statistical approach is interspersed throughout the ‘Methods’ and ‘Results’ sections and should be in a separate ‘Statistical Analysis’ section describing the overall analysis approach.

6. PLOS authors have the option to publish the peer review history of their article (what does this mean?). If published, this will include your full peer review and any attached files.

Reviewer #1: No

Reviewer #2: No

---

## [Author Response · Author response to Decision Letter 0]

23 Nov 2020

Comments from the editor

Q1. Answer: Revised and added separate para. to explain at pg.4, 7 and 8.

Q2. Answer: A power test is run and explain on p. 9

Q3. Answer: Done

Q4. Answer: Done

Q5. Answer: Done

Q6. Answer: Done

Q7. Answer: Done

Reviewers' comments to Authors

Q5 from Reviewer #1: 

Answer: 

Revised and added one more para. at pg.4 to discuss more about cross-cultural differences in spirituality. 

Revised and add one more section to explain the rationale and the history at pg.8.

Revised and add one sentence section to discuss the responds at pg.7. 

Q6 from Reviewer #2

Answer: Revised and two separate para. to discuss the quantitative findings and statistical analysis respectively

---

## [Decision Letter · Decision Letter 1]

8 Dec 2020

PONE-D-20-25063R1

Cross-Sectional Study of the Relationship between the Spiritual Wellbeing and Psychological Health among University Students

PLOS ONE

Dear Dr. Chi Hung Leung and co-workers,

Thank you for submitting your manuscript to PLOS ONE. After careful consideration, we feel that it has merit but does not fully meet PLOS ONE’s publication criteria as it currently stands. Therefore, we invite you to submit a revised version of the manuscript that addresses the points raised during the review process.

Please do pay full attention to the statistical shortcoming of your study design and the overall style and grammar of the mansucript.

We look forward to receiving your revised manuscript.

Kind regards,

Jacobus P. van Wouwe, MD PhD

Academic Editor

PLOS ONE

Additional Editor Comments (if provided):

Please pay proper attention toward the suggestion of a power analysis. This is a major shortcoming of your study that needs to be corrected. In a proper power analysis you need to calculate how valid the difference in the endpoints of your study need to be in order to draw valid conclusions.

Secondly, please notice the awkward distinction you seem to make between Christians and Catholics. This seems to be due to the errors in style and grammar. Please consult a professional review service.

Reviewers' comments:

Reviewer's Responses to Questions

**Comments to the Author**

1. If the authors have adequately addressed your comments raised in a previous round of review and you feel that this manuscript is now acceptable for publication, you may indicate that here to bypass the “Comments to the Author” section, enter your conflict of interest statement in the “Confidential to Editor” section, and submit your "Accept" recommendation.

Reviewer #1: (No Response)

Reviewer #2: (No Response)

2. Is the manuscript technically sound, and do the data support the conclusions?

Reviewer #1: Yes

Reviewer #2: Partly

3. Has the statistical analysis been performed appropriately and rigorously? 

Reviewer #1: Yes

Reviewer #2: No

4. Have the authors made all data underlying the findings in their manuscript fully available?

Reviewer #1: Yes

Reviewer #2: Yes

5. Is the manuscript presented in an intelligible fashion and written in standard English?

Reviewer #1: No

Reviewer #2: Yes

6. Review Comments to the Author

Reviewer #1: The paper is much improved. The addition of material on cross-cultural differences in spirituality is much appreciated. It helps give context.

Despite these improvements, there are still some necessary but less major changes that must be made:

1) There needs to be better labeling of religious groups (this is more about labels than the actual data analysis).

The paper distinguishes between "Catholics" and "Christians." This is a mistake in labeling because Catholicism is a form of Christianity. A better distinction would be between "Catholics" and "Other Christians."

This change would need to be made:

-Top of page 9 change "Christians" to "Other Christians" in both places

-Change "Christians" to "Other Christians" in the results on page 13 in Table 2

-Change "Christians" to "Other Christians" on page 17

2) There are still some writing mistakes that need to be corrected (this is especially important because the journal does not do copy editing):

-"Potential responds to the campus mental health crisis" should be changed to "Potential response to the campus mental health crisis" (change "responds" to "response")

-This sentence on page 18 is not grammatically correct: In the Asia-Pacific region, and it is the only study to contribute the links

between spiritual wellbeing and psychological disorders among university students.

It's possible that an extra "and" was inserted in the sentence. This may be the correct wording:

"In the Asia-Pacific region, it is the only study to contribute the links

between spiritual wellbeing and psychological disorders among university students."

I would urge the authors to proofread the manuscript for any grammar or writing mistakes (or solicit the services of a seasoned editor at their institution(s)).

Despite these minor changes, I believe this paper is ready for publication and would be a solid contribution to the literature. The discussion of spirituality and mental health on campus is very very timely and this paper presents some very relevant findings.

Reviewer #2: The authors address the power question in a very cavalier manner. They note that a power analysis of the sample size with a confidence level of 95% is 383 [99]. A sample of 500 Chinese university students is enough for the present study. This does not make any sense. What endpoint objectives were used in this calculation ?

7. PLOS authors have the option to publish the peer review history of their article (what does this mean?). If published, this will include your full peer review and any attached files.

Reviewer #1: No

Reviewer #2: No

---

## [Author Response · Author response to Decision Letter 1]

8 Jan 2021

Dear Editors, 

We thank the reviewers for their generous comments on the manuscript. We have edited the manuscript again to address their concerns. We believe that the manuscript is now suitable for publication in PLOS ONE. The responses to the comments are shown below: 

 Comments from the reviewer 1 Responses to the comments

1. 1) There needs to be better labeling of religious groups (this is more about labels than the actual data analysis).

The paper distinguishes between "Catholics" and "Christians." This is a mistake in labeling because Catholicism is a form of Christianity. A better distinction would be between "Catholics" and "Other Christians."

This change would need to be made:

-Top of page 9 change "Christians" to "Other Christians" in both places

-Change "Christians" to "Other Christians" in the results on page 13 in Table 2

-Change "Christians" to "Other Christians" on page 17 

Revised in Page 9, 13 and 17. 

2. 2) There are still some writing mistakes that need to be corrected (this is especially important because the journal does not do copy editing):

-"Potential responds to the campus mental health crisis" should be changed to "Potential response to the campus mental health crisis" (change "responds" to "response")

-This sentence on page 18 is not grammatically correct: In the Asia-Pacific region, and it is the only study to contribute the links

between spiritual wellbeing and psychological disorders among university students.

It's possible that an extra "and" was inserted in the sentence. This may be the correct wording:

"In the Asia-Pacific region, it is the only study to contribute the links

between spiritual wellbeing and psychological disorders among university students."

Revised in page 7. 

Revised in page 18. 

3. I would urge the authors to proofread the manuscript for any grammar or writing mistakes (or solicit the services of a seasoned editor at their institution(s)). Revised in Page ….. (write down the page no. after proofreading)

 Comments from the reviewer 2 Responses to the comments

1. Please pay proper attention toward the suggestion of a power analysis. This is a major shortcoming of your study that needs to be corrected. In a proper power analysis you need to calculate how valid the difference in the endpoints of your study need to be in order to draw valid conclusions. Further explain in page 8. 

‘The result of the power indicated a very precise estimate of the regression coefficient with a very small confidence interval around it and the representativeness of the sample.’

Sincerely, 

Prof. Leung Chi Hung,

---

## [Decision Letter · Decision Letter 2]

16 Mar 2021

PONE-D-20-25063R2

Cross-Sectional Study of the Relationship between the Spiritual Wellbeing and Psychological Health among University Students

PLOS ONE

Dear Dr. Chi Hung Leung and co-workers,

Thank you for submitting your manuscript to PLOS ONE. After careful consideration, we feel that it has merit but does not fully meet PLOS ONE’s publication criteria as it currently stands. Therefore, we invite you to submit a revised version of the manuscript that addresses the points raised during the review process.

We have given the statistics thorough attention and hope you agree on the suggestions for clarification. We are sorry it took longer.

We look forward to receiving your revised manuscript.

Kind regards,

Jacobus P. van Wouwe, MD PhD

Academic Editor

PLOS ONE

Journal Requirements:

Additional Editor Comments (if provided):

The statistical review has been thorough and please do follow these seven minor additional clarifications. Thank you.

Reviewers' comments:

Reviewer's Responses to Questions

**Comments to the Author**

1. If the authors have adequately addressed your comments raised in a previous round of review and you feel that this manuscript is now acceptable for publication, you may indicate that here to bypass the “Comments to the Author” section, enter your conflict of interest statement in the “Confidential to Editor” section, and submit your "Accept" recommendation.

Reviewer #3: (No Response)

2. Is the manuscript technically sound, and do the data support the conclusions?

Reviewer #3: Yes

3. Has the statistical analysis been performed appropriately and rigorously? 

Reviewer #3: Yes

4. Have the authors made all data underlying the findings in their manuscript fully available?

Reviewer #3: Yes

5. Is the manuscript presented in an intelligible fashion and written in standard English?

Reviewer #3: Yes

6. Review Comments to the Author

Reviewer #3: This observational study tested the relationship between spiritual wellbeing and symptoms of psychological disorders (i.e., depression, anxiety and stress) among Chinese university students in Hong Kong. All domains of spiritual well being were negatively associated with psychological distress.

Minor revisions:

1- Words or two seems to be missing in the first paragraph under “Participants,” line begins with “median.”

2- In the paragraph beginning with, “A series of t-tests,” the statement indicates that no differences were observed but p-values were < 0.01. Clarify.

3- Consider categorizing age in the analyses.

4- In Table 3: Replace “Number” with frequency for clarity. Add formats or reform the table to easily distinguish between Depression, Anxiety, and Stress.

5- The p-value associated with a correlation is a test of the null hypothesis: correlation equal to zero; however, the absolute magnitude of the coefficient indicates the strength of the linear relationship between two variables. In general, the strength or correlation coefficient is the more important statistic to reflect upon.

Below is a table for interpreting correlation coefficients:

Coefficient (absolute value) Interpretation

0.90 - 1.0 Very Strong

0.70 - 0.89 Strong

0.40 - 0.69 Moderate

0.10 - 0.39 Weak

less than 0.10 Negligible correlation

6- Define all abbreviations on Table 5.

7. PLOS authors have the option to publish the peer review history of their article (what does this mean?). If published, this will include your full peer review and any attached files.

Reviewer #3: No

---

## [Author Response · Author response to Decision Letter 2]

21 Mar 2021

Comments from the reviewer 3 Responses to the comments

1. Minor revisions:

1- Words or two seems to be missing in the first paragraph under “Participants,” line begins with “median.”

2- In the paragraph beginning with, “A series of t-tests,” the statement indicates that no differences were observed but p-values were < 0.01. Clarify.

3- Consider categorizing age in the analyses.

4- In Table 3: Replace “Number” with frequency for clarity. Add formats or reform the table to easily distinguish between Depression, Anxiety, and Stress.

5- The p-value associated with a correlation is a test of the null hypothesis: correlation equal to zero; however, the absolute magnitude of the coefficient indicates the strength of the linear relationship between two variables. In general, the strength or correlation coefficient is the more important statistic to reflect upon.

Below is a table for interpreting correlation coefficients:

Coefficient (absolute value) Interpretation

0.90 - 1.0 Very Strong

0.70 - 0.89 Strong

0.40 - 0.69 Moderate

0.10 - 0.39 Weak

less than 0.10 Negligible correlation

6- Define all abbreviations on Table 5. 1. We may not really know where the “median” is. Would you mind highlighting the sentence that I have to amend? Thanks a lot. 

2. fixed p > .05

3. fixed 17 – 19, 20, 21 – 23

 (see Table 2)

4. Number has been replaced by frequency and table has also been reformatted

5. Thank you, the magnitude of the coefficient has been edited based on the table provided.

6. all abbs have been defined

---

## [Editor Report · Decision Letter 3]

24 Mar 2021

Cross-Sectional Study of the Relationship between the Spiritual Wellbeing and Psychological Health among University Students

PONE-D-20-25063R3

Dear Prof Dr. Chi Hung Leung and co-workers,

We’re pleased to inform you that your manuscript has been judged scientifically suitable for publication and will be formally accepted for publication once it meets all outstanding technical requirements. The two changes above mentioned as suggested during the proofs (to delete 'very' twice) will be greatly appreciated.

Kind regards,

Jacobus P. van Wouwe, MD PhD

Academic Editor

PLOS ONE

Additional Editor Comments (optional):

Thank you for responding to the reviewer's comments.

May I suggest you delete under Methods, Participants, the following two words:

'very precise' change to 'precise' and 'very small' change to 'small'. It would be greatly appreciated. Thank you.
---

## [Editor Report · Acceptance letter]

5 Apr 2021

PONE-D-20-25063R3 

Cross-Sectional Study of the Relationship between the Spiritual Wellbeing and Psychological Health among University Students 

Dear Dr. Leung:

I'm pleased to inform you that your manuscript has been deemed suitable for publication in PLOS ONE. Congratulations! Your manuscript is now with our production department. 

Kind regards, 

on behalf of

Dr. Jacobus P. van Wouwe 

Academic Editor

PLOS ONE